# Chemical Vapor Deposition of Longitudinal Homogeneous Parylene Thin-Films inside Narrow Tubes

**David Redka ***[ID]**, Milan Buttberg and Gerhard Franz** [ID]

Department of Applied Sciences and Mechatronics, Munich University of Applied Sciences HM, Lothstr. 34, 80335 Munich, Germany

* Correspondence: dredka@hm.edu

**Abstract:** The effect of quasi-exponentially decreasing film thicknesses of thin poly-*para*-xylylene (PPX-N) coatings inside narrow tubes or micro scaled gaps is well known and has been discussed by many authors since the late 1970s. However, for technical applications it is often necessary to provide a longitudinal homogeneous film thickness to ensure the constant properties that are required. In a previous work, it was shown, in principle and for the first time, that a temperature gradient along the tube will effectively counteract the longitudinal decreasing film thickness of the PPX-N coating of the interior wall of a capillary. Therefore, this effect is discussed in theory and the provided model is verified by experiments. Our prediction of a required sticking coefficient curve yields experimentally measured homogeneous film thicknesses and shows a good agreement with the given prognosis. Further, it is shown in theory that there is a maximum achievable homogeneous film thickness in the tube in comparison to a blank surface, which can be understood as a coating efficiency for this type of deposition.

**Keywords:** parylene; chemical vapor deposition; deposition model; constant film thickness; sticking coefficient; temperature gradient

## 1. Introduction

Chemical vapor deposition (CVD) is well known for forming high conformal coatings with uniform thickness on rugged surfaces with deep valleys or high mountains, on nearly all types of materials. Especially for modern applications as in medical [1–5] or in electronic devices [6–8], where constant film properties across the whole substrate are demanded, this type of processing has nowadays found widespread usage.

In contrast to other deposition techniques, e.g., sputtering or evaporation techniques, which involve atoms as film-building species, reactive molecules or radicals in CVD form a polymeric film by a chemical reaction. According to Arrhenius and the law of mass action, chemical reactions always depend strongly on temperature [9,10], and their manufacture is challenge with respect to process parameters and their stability [11–13].

For the coating of medical implants, one of the most promising polymers is poly-*para*-xylylene (PPX-N), or other related derivatives [14,15], providing the ability to form high-precision thin-films on substrates. These thin-films have excellent properties such as transparency, high temperature resistance, mechanical flexibility, bio-compatibility, and chemical resistance and can even act as diffusion barriers [8,16,17]. Hereby, various methods are known to influence the film quality, properties and growth rate during PPX-N CVD. For example, diluting the monomer vapor with a non-reactive residual gas, such as argon, reduces the film growth rate but yields a higher optical transparency [18], and is therefore a state of the art technique in industry and research.

Thin films of PPX-N are typically formed within the so-called Gorham process [19], where a precursor is evaporated under reduced pressure (temperatures between 120 °C and 150 °C, total pressure in the order of 10 Pa). The dimeric species is thermally cracked

into two di-radical monomers according to the simple sum reaction $DPX \rightarrow 2\,MPX$, which flow through a monomer distributor into the reaction chamber.

In principle, the process of polymerization is known to be a surface phenomenon [20], where the monomer particles are adsorbed and subsequently chemisorbed with a specific probability, so a polymerization on the substrate but also on the walls of the whole reaction chamber takes place [21]. Furthermore, the polymerization is determined to be kinetically controlled [10], which means that the film growth rate is proportional to the reactants surface concentration in the vapor phase (local partial pressure of the monomer) and the reaction kinetics during the polymerization, which are known to show a dependence on the surface temperature [22–24]. Experimentally, all authors have found a decrease of growth rate with an increase of temperature. Eventually, at the so-called ceiling temperature, no deposition is possible. The ceiling temperature for PPX-N has been communicated to be between 25 °C [23–25] and 68 °C [20].

In the past decades, different models for the deposition kinetics of PPX-N have been introduced, predicting the film growth rate of the polymer in dependence of the surface temperature of the substrate $T_s$ and the reactants concentration in the gas phase, as well as the partial pressure of the di-radical monomer. In short, these models show a dependence on the partial pressure $p_m$ to be proportional to $p_m^n$ with values for $n$ between one and two [22,23,25], where the linear model seems to represent the best compromise between complexity and predicting experiments [23]. In contrast to CVD processes which form inorganic films such as $SiO_2$, these models also predict a decreasing deposition rate with rising surface temperature which eventually vanishes.

Despite its advantages of forming high conformal coatings, a homogeneous film thickness along an extended substrate can hardly be reached using CVD. This fact is caused by the inevitable depletion of the reactant with growing distance from the evaporating source, leading to a decreasing film thickness along the whole reaction chamber [19]. This decrease of the film thickness is even more pronounced along the interior wall of tubes, capillaries or micro scaled gaps. In the past decades, various models explaining the well-known exponential film thickness decrease inside narrow tubes have been developed [7,26–28]. Usually, the authors consider a one-dimensional axis-symmetric gas flow through the tube for different types of gas–dynamic regimes. These models all predict with some accuracy the exponential decrease in film thickness along a capillary, but none treats the possibility of manipulating the deposition behavior. Moreover, a detailed investigation of the slip–flow regime, where the monomer mean free path is comparable to the geometry of the system, here the diameter of the tube, is still missing in this context. However, a detailed description is important since, at characteristic process pressures of several Pa, the two case limiting theories, namely molecular flow (e.g., microscale gaps) and viscous flow (e.g., flow in the reactor), break down for promising applications such as in medicine (stents, catheters, and further [20,21]).

In order to counteract the effect of decreasing film thicknesses, and, further, to produce longitudinally homogeneous coatings inside narrow tubes by means of CVD, we have already shown in a previous work that spatial manipulation of the sticking coefficient, by application of a thermal gradient, yields reasonable results [20]. However, a theoretical approach to understand this process is still missing, especially in the slip–flow regime where the mean free path of the film-building species is similar to the cross section of the capillary. Since a direct access on the PPX-N partial pressure along the tube is in principle denied, our proposed method seems to be a unique solution tackling this problem. We further expect our method to be applicable for related processes.

In Section 2, we discuss our approach in a more formal basis and present a detailed investigation of the CVD deposition of PPX-N inside narrow tubes. Section 3 gives an overview on the experimental procedure in this work. In Section 4, first the results on the deposition in narrow tubes at constant temperatures are discussed, followed by measurements of the temperature-dependent sticking coefficient. With knowledge on our system, we are further able to derive a required sticking coefficient gradient in order to reach a

homogeneous film thickness inside narrow tubes. Finally, with this prediction, we are in a position to generate high quality homogeneous film thicknesses along narrow tubes and validate the proposed model by comparison of simulation and experiments.

## 2. Theory on PPX-N CVD in Narrow Tubes

### 2.1. General Model for the Deposition in Narrow Tubes

The film thickness growth of PPX is generally understood as surface polymerization of di-radical monomers from the gas phase upon contact with a surface [19,22,25,29]. Initial models are based on the adsorption of monomers on the surface with Flory-type [30] or Langmuir-type [22] adsorption isotherms. Subsequently, a reaction kinetic separation with partial temperature-dependent diffusion of monomers to the next reactant in the forming film (chain end) as well as an additional surface polymerization was considered [22]. Due to the complex modeling, there are different proportionalities in the growth rate of the film thickness, such as those from the monomer partial pressure in the adsorption isotherms or the substrate temperature, which exert influence on the diffusion coefficient as well as the vapor pressure of the monomer during Flory adsorption [22]. However, these models have the disadvantage of consisting of many complex terms with free parameters that may contain effective terms and are difficult to study separately [10]. At this point, a chemisorption model was proposed by Fortin et al. [23] based on a work investigating the sticking probability of chemisorbed gases on surfaces [31]. Here, the equilibrium rate between migration, desorption, and chemisorption of adsorbed molecules was combined into a generalized sticking coefficient [10,23]. Temperature dependence was addressed by the Boltzmann factor within the reaction probabilities, introducing a dependence on surface temperature within this model.

By defining a sticking coefficient for the di-radical parylene monomer, the film-growth rate of PPX-N is given by the product of molecules from the gas phase that chemisorb after collision with the surface, (with $j_M$ the flux density and $S(T_s)$ the sticking coefficient in dependence of the substrate surface temperature $T_s$) and the molecular volume $V_M$ of the reactant. This relation is in dependence of the partial pressure of the film forming monomer $p_M$ written as

$$\dot{d}_f = j_M \, V_M \, S(T_s) = \frac{1}{\sqrt{\pi} \, \hat{c}_g \, \rho_f} \, S(T_s) \, p_M \tag{1}$$

Here $\hat{c}_g$ is the most probable velocity of the di-radical monomer from kinetic gas theory and $\rho_f$ the film density. Herewith, a mathematical expression for the film thickness, in dependence of a spatial manipulated sticking coefficient, along the interior wall of a one-sided closed tube can be derived. We assume the CVD conditions as steady state, which means that the timescale of the change on monomer pressure at the inlet of the tube (due to possible effects of process-related pressure changes) is much greater than the diffusion time of the monomer from the inlet of the tube to its end. In addition, enough particle collisions of initial hot monomers and the residual gas or the reactors walls, both at room temperature, are given between cracker and substrate, so that the molecules will be in thermal equilibrium with the room temperature at the deposition site [32–34]. Due to the choice of a rotation-symmetrical tube, the problem may be reduced towards one dimension (parallel to tube axis). Neglecting the transient change of the tube's radius $R_t$ caused by the film growth, the change of monomer mass flow per distance interval inside the tube is in combination with Equation (1) given by

$$-\frac{d\dot{m}_M}{dx} = 2 \, \pi \, R_t \, \rho_f \, \dot{d}_f = \frac{2 \, \sqrt{\pi} \, R_t}{\hat{c}_M} \, S(T_s) \, p_M \tag{2}$$

On the other hand, a pressure or temperature gradient of a gas, with the local gas temperature $T_g$—here the gaseous monomer, in a tube will induce a local mass transfer, which is for the investigated cylindrical geometry given by [35]

$$\dot{m}_M = \frac{\pi R_t^3}{\hat{c}_M}\, p_M \left( \frac{G_T}{T_g} \frac{dT_g}{dx} - \frac{G_p}{p_M} \frac{dp_M}{dx} \right) \tag{3}$$

Here $G_p$ is referred to the Poiseuille coefficient and the term containing $G_T$ describes the induced gas flow by a relative temperature gradient. With Equations (2) and (3), the ordinary differential equation for our problem is given as

$$\frac{S(T_s)}{\hat{c}_M}\, p_M = \frac{\sqrt{\pi}\, R_t^2}{2} \frac{d}{dx}\left[ \frac{p_M}{\hat{c}_g} \left( \frac{G_p}{p_M} \frac{dp_M}{dx} - \frac{G_T}{T_g} \frac{dT_g}{dx} \right) \right] \tag{4}$$

The boundary condition at the closed end of the tube $x = L$ is given by the film thickness growth on the perpendicular closing tube surface $A_t$ induced by the mass flow of the reactant at this position $\dot{m}_M(L) = \rho_f\, \dot{d}_f(L)\, A_t$, and is (with Equations (1) and (3))

$$\left. \frac{dp_M}{dx} \right|_L = -\frac{1}{\sqrt{\pi}\, R_t} \frac{S(L)}{G_p(L)}\, p_M(L) \tag{5}$$

Thereby, the temperature gradient at the end of the tube is set to zero. At the inlet, the boundary value is given by the partial pressure of the reactant in the gas phase $p_M(0)$. By introducing the relative variables $P = p_M/p_M(0)$ and $X = x/L$, the boundary value problem is rewritten in a more general form

$$\frac{S(T_s)}{\hat{c}_g}\, P = \frac{\sqrt{\pi}}{2} \left( \frac{R_t}{L_t} \right)^2 \frac{d}{dX}\left[ \frac{P}{\hat{c}_g} \left( \frac{G_p}{p_M} \frac{dP}{dX} - \frac{G_T}{T_g} \frac{dT_g}{dX} \right) \right],\; \left. \frac{dP}{dX} \right|_1 = -\frac{1}{\sqrt{\pi}} \frac{L_t}{R_t} \frac{S(1)}{G_p(1)} P(1),\; P(0) = 1 \tag{6}$$

In order determine the film thickness profile, the solution of Equation (6) is inserted back into Equation (1) and integrated over the process duration $\tau$ taking into account a slowly varying (much less than the diffusion time) Parylene partial pressure at the inlet

$$d_F(X, \tau) = \frac{S(X)}{\sqrt{\pi}\, \rho_f\, \hat{c}_g(X)} \int_0^\tau dt\, p_M(0, t)\, P(X, t) \tag{7}$$

where for comparison of experiment and theory the relative film-thickness with respect to the inlet may be used [5,6]

$$D(X) = d_f(X, \tau)/d_f(0, \tau) \tag{8}$$

It is apparent that the complex form of Equation (6) is not analytically solvable for a spatial arbitrary sticking coefficient. The coefficients $G_p$ and $G_T$ are also dependent on the gas rarefaction parameter $\delta = 2\, R_t/\lambda$ [35] and the monomer partial pressure $p_M$ and respectively the $X$ coordinate. Therefore, an approximation for our case will be presented in the following subsection.

However, prior a brief comparison to existing literature models is carried out. A common thread among these models is that solely steady-state conditions and a spatial constant sticking coefficient ($p_M(0, t) = p_M(0)$ and $S(X) = S$) are taken into account. In addition, common approximations are constant Poiseuille coefficients or diffusion coefficients, respectively. According to the given assumptions the boundary value problem of Equation (6) simplifies to

$$\frac{d^2 P}{dX^2} = \frac{2}{\sqrt{\pi}} \frac{L_t^2}{R_t^2} \frac{S(T_s)}{\langle G_p \rangle}\, P = \alpha\, P,\; P(0) = 1,\; \left. \frac{dP}{dX} \right|_1 = -\frac{\alpha}{2} \frac{R_t}{L_t} P(1) \tag{9}$$

Hereby all constants are set into one loss coefficient $\alpha$. The result of $P$ is well known and was also reported by Tolstopyatov et al. [28]. Also from Equation (8) it becomes

apparent that the relative pressure for these conditions is equivalent to the relative film thickness which yields

$$P(X) = D(X) = \frac{2\,L_t \cosh\left(\sqrt{\alpha}\,(1-X)\right) + R_t\,\sqrt{\alpha}\sinh\left(\sqrt{\alpha}\,(1-X)\right)}{2\,L_t \cosh\left(\sqrt{\alpha}\,\right) + R_t\,\sqrt{\alpha}\sinh\left(\sqrt{\alpha}\right)} \tag{10}$$

For high aspect ratios, $Ar = L_t/R_t$, and an intermediate loss coefficient, namely $(L_t/R_t)^2 \gg \alpha/4$ this simplifies to [27]

$$P(X) = D(X) = \frac{\cosh\left(\sqrt{\alpha}\,(1-X)\right)}{\cosh\left(\sqrt{\alpha}\,\right)} \tag{11}$$

which is equivalent to neglecting the film growth at the tubes closed end, mathematically described by $\left.\frac{dP}{dX}\right|_1 = 0$.

### 2.2. PPX-N Deposition in the Slip-Flow Regime

In our case of PPX-N, CVD we diluted the reactant with a residual noble gas, which is in common use to ensure a more controllable deposition [18]. By knowledge of both partial pressures, the mean free path $\lambda$ of the monomer in the residual gas is [36]

$$\lambda = \frac{k_B T_g}{\sqrt{2}\,\sigma_{MM}\,p_M + \sqrt{1 + M_M/M_R}\,\sigma_{MR}\,p_R} = \frac{k_B T_g}{\sigma_1\,p_M + \sigma_2\,p_R} \tag{12}$$

with $T_g$ the gas temperature and $p_{M,R}$ the partial pressure of the monomer and residual gas, respectively. In this work we use argon as diluting gas. The molar masses are $M_M \approx 104\ \text{g/mol}$ and $M_R \approx 40\ \text{g/mol}$, and the geometric cross section $\sigma_{MM} \approx 634\ \text{Å}^2$ is approximated from the van der Waals surface area of *para*-xylylene [37]. The mixed geometric cross section $\sigma_{MR} = \pi\,(r_M + r_R)^2 \approx 253\ \text{Å}^2$ is calculated by assuming the monomer cross section to be circular. With 10 Pa argon base pressure and a monomer partial pressure of around 3 Pa the rarefaction parameter is $\delta = 2\,R_t/\lambda \approx 20\ldots40$ at the inlet of the tube which has an inner diameter of $R_t = 1\ \text{mm}$. Therefore, the gas dynamic regime is characterized to be in the slip–flow regime [35], and $G_p$ and $G_T$ are approximated from molecular dynamic simulations by [38,39]

$$G_p \approx 1 + \frac{\delta}{4}\,,\ G_T \approx \frac{1}{\delta} \tag{13}$$

It follows that in the slip–flow regime, the magnitude of $G_p$ is in the order of two decades above $G_T$ ($G_p \gg G_T$) and therefore the term with the temperature slip coefficient $G_T$ in (5) can be neglected if the relative temperature and pressure gradients are in the same order of magnitude. This is the case for our manipulation, with the sticking coefficient and pressure being linear proportional to the deposition rate (see Equation (1)). Equation (6) is rewritten with $d\hat{c}_g/dX = (\hat{c}_g/2T_g)(dT_g/dX)$ as

$$\frac{d^2P}{dX^2} + \left[\frac{1}{G_p}\frac{dG_p}{dX} - \frac{1}{2\,T_g}\frac{dT_g}{dX}\right]\frac{dP}{dX} - \frac{2}{\sqrt{\pi}}\left(\frac{L_t}{R_t}\right)^2\frac{S(T_s)}{G_p}\,P = 0,\ \left.\frac{dP}{dX}\right|_1 = -\frac{1}{\sqrt{\pi}}\frac{L_t}{R_t}\frac{S(1)}{G_p(1)}P(1),\ P(0) = 1 \tag{14}$$

For intermediate dilute gas mixtures, the total gas pressure will stay constant over the whole tube distance $X$. Thus a rarefaction in the monomer partial pressure will increase the partial pressure of the residual gas [26] given by

$$\frac{d}{dX}(p_M + p_R) = 0 \rightarrow p_{tot} = p_M(X) + p_R(X) \tag{15}$$

where $p_{tot}$ is the total pressure at the inlet of the tube and can be experimentally tracked during the deposition process. With this information, the Poiseuille coefficient is expressed

only showing dependence in the partial pressure of the monomer and its temperature. Due to the condition from Equation (15), $G_p$ and its derivative with respect to the space variable can be expressed via Equations (12) and (13) as

$$G_p(X) = 1 + \frac{R_t}{2k_B T_g} \left[ (\sigma_1 - \sigma_2)\, p_M + \sigma_2\, p_{tot} \right] \qquad (16)$$

At this point it should be noted that the gas temperature inside the tube being influenced by the surface temperature should be discussed. Although the slip–flow regime is characterized by a temperature jump between the vacuum and the wall [40], radial effects are not considered in our 1D approach. Therefore, in Section 4 both cases with $T_g = T_s$ and $T_g = T_{ref}$ will be examined.

## 3. Materials and Methods

CVD experiments were performed in an in-house constructed deposition reactor (see Figure 1a), consisting of an evaporation chamber, a thermal cracker unit (heated by infrared radiation at 600 °C [41]), a monomer distributor (100 °C, above $T_c$ to avoid film growth) and a flow-through reaction chamber. The corresponding temperatures during the deposition process were set by a PID controller. The pressure in the reaction chamber was measured with a heated (90 °C, above $T_c$) capacitance manometer. The mass flow of the diluent gas argon was set by a mass-flow controller with a mass flow of approximately 8.5 sccm resulting in 10 Pa base pressure. The argon gas inlet to the reactor was located between the cracker and the monomer distributor (see Figure 1a), ensuring a homogeneous mixture of both gas species. This allows the hot MPX to be cooled down in thermodynamic equilibrium to near room temperature before reaching the reaction chamber. For the whole reactor the gas dynamic regime is in the viscous flow within our pressure range. Therefore, the MPX partial pressure decrease in the reaction chamber due to surface polymerization on its walls is nearly negligible due to the low aspect ratio of close to one (see below). The PPX-N precursor was provided by Plasma Parylene Systems (Rosenheim, Germany). To ensure comparability, relevant process parameters were kept constant during all experiments. In order to suppress evaporation during the heating, the argon pressure was set to 100 Pa (130 sccm) and subsequently controlled with the mass flow controller to a base pressure of 10 Pa, where the deposition process was initiated. A precursor mass of 5.0 g was evaporated at a temperature of 115 °C. The capacitance manometer was placed as close as possible to the substrate, respectively inlet of the tubes, to best represent the concentration of the monomer in the gas phase above the substrate (see Figure 1a). The partial pressure of the monomer was determined from the measured total pressure, as the relation of argon mass flow and pressure is known for the reactor. However, since the precursor mass is limited and the open evaporator unit refers to a non-thermodynamic equilibrium system, and therefore cannot maintain a stable vapor pressure, the PPX-N mass flow and, respectively, partial pressure were not constant over time during the experiments. As the evaporation rate is proportional to the existing precursor surface, and this in turn to the partial pressure, an exponential progression of the PPX monomer partial pressure is expected (($P_M(t) \propto \exp(-\gamma t)$, with $\gamma$ the evaporation rate constant). A typical pressure trend is depicted in Figure 1d. As can be seen, small deviations in the evaporator temperature control unit ($\Delta T \approx 2$ °C) lead to minor oscillations in the pressure. A coating thickness of 2.6(3) μm on a $T_{ref} = 30$ °C heated soda lime glass, positioned in the center of the reaction chamber, was measured for all experiments.

In order to determine the temperature dependence of the sticking coefficient, an aluminum slab attached to a resistive heating and water-cooled unit was placed with thermally connected soda lime glass microscope slides inside the reactor chamber. The setup is shown in Figure 1b. Due to the constant thermal flow within the aluminum slab, a linear temperature gradient is established on the microscope slides. The temperature was measured at two points with K-type thermocouple elements and may be interpolated for further analysis. The rarefaction of the reactant, caused by the deposition on the reactor

walls itself, was observed and taken into account by measuring the film thickness along the microscope slides for a coating process where the temperature was held constant at 30 °C. Due to the low aspect ratio (length over diameter) of the reaction chamber, the film growth rate along the two microscope slides (in total, approx. 17 cm) was expected to remain almost constant. In fact, the measurements showed that the relative film thickness changes quasi-linearly in the range of 2%, which was taken into account in the evaluation of the temperature dependence of the sticking coefficient.

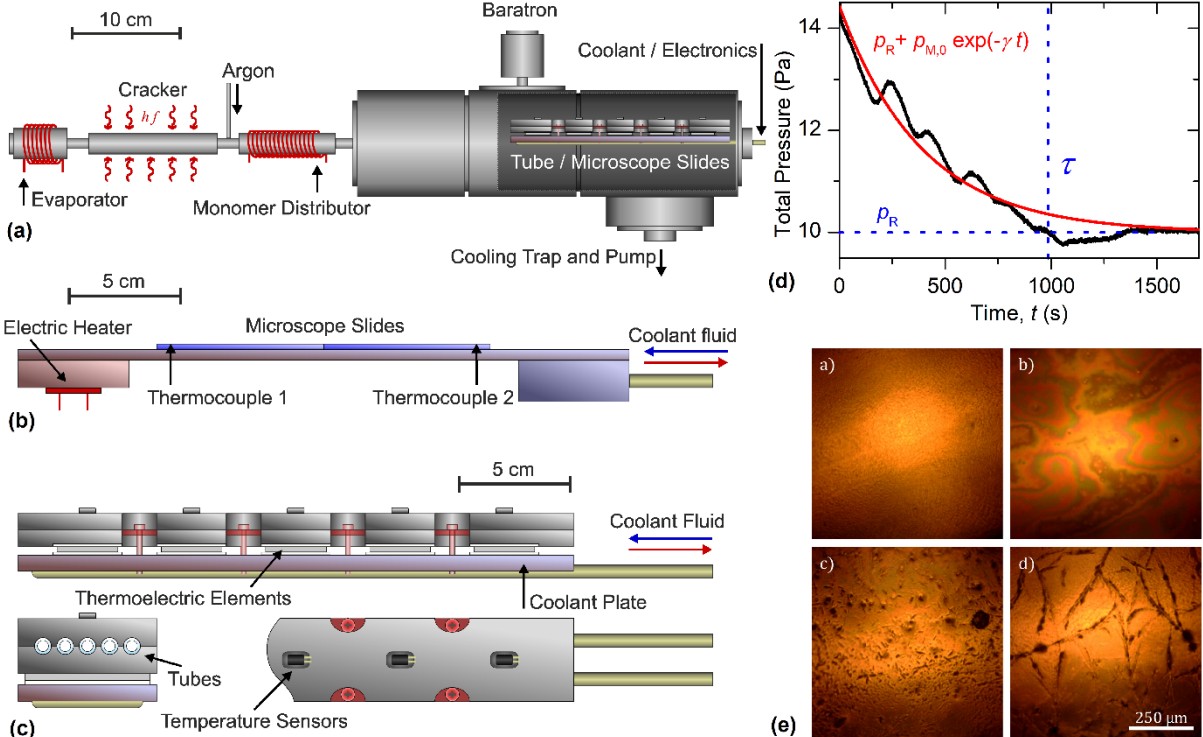

**Figure 1.** (**a**) Sketch of the experimental system containing of a heated evaporator, a thermal cracker and a monomer distributor unit. The experimental setup was placed inside the reaction chamber and the pressure was measured with a capacitance manometer (baratron); (**b**) setup to determine the temperature dependence of the sticking coefficient, containing an aluminum slab with a constant thermal flow. The position of the temperature measurement points and microscope slides is indicated; (**c**) temperature seesaw enables the spatial manipulation of the sticking coefficient inside narrow tubes. The tubes were clamped between two rails, with the temperature control being obtained by five thermoelectric elements and temperature sensors; (**d**) typical total pressure versus time for the deposition process measured with the baratron close to the deposition sample. The quasi exponential decay of the MPX partial pressure is indicated as well the argon base pressure; (**e**) microscopy image of the deposited film inside a tube for various treatments after the deposition with (a) smooth and intact deposited PPX layer, (b) warped layer with interferences, (c,d) cracked and warped thin film at around 5 cm due to intrinsic tension. This effect can be avoided by applying a constant temperature after the deposition.

For our interior tube coatings we used one-sided closed polyamide (PA) tubes with a length of 20 cm, an inner diameter of 2 mm and wall thickness of 1 mm. The tubes were placed inside an advanced version of an experimental setup we introduced in a previous work, called temperature seesaw [20] (see Figure 1c). This device consists of two steel rails with cutouts for up to five tubes, which are clamped between the lower and upper rails. By mounting five thermoelectric elements on the bottom, virtually any temperature gradient, with a corresponding degree of accuracy, can be applied to the contact surface of the polyamide tubes. Temperature control was realized by means of a PID controller and five temperature sensors mounted on the top of the upper rail. A constant temperature at

the lower contact surface of the thermoelectric elements was maintained using an external chiller. Local temperatures (measured with five thermocouples on the top of the upper rail) were captured digitally during the deposition process. The temporal deviation was less than a half degree Celsius in all experiments.

In order to determine the progression of the temperature gradient of the tubes inner surface, the steady-state heat equation was solved, with the measured and respectively set temperatures at the reference points as boundary value conditions. The commercial software COMSOL Multiphysics 5.5 (Heat Transport Module) was used for this purpose. The thermoelectric elements were replaced by open heat flow boundary conditions to decrease the complexity of the system. A thermal resistance layer was introduced between the upper and lower rail in order to take a possible temperature jump at the contact surface into account. The solution for the temperature field was used to determine the local sticking coefficient and finally to reconcile the numerical solution of Equation (14) and experiments. The boundary value problem, given in Section 2 was solved using COMSOL Multiphysics 5.5 (Coefficient Form PDE).

After all experiments, the deposited local PPX-N coating thickness was measured by optical thin-film interferometry (F20, Filmetrics, Unterhaching, Germany). Here, the acquired measurement spot was about 30 μm, which made it possible to measure not only the coated slides but also the film thickness inside the tubes after they were cut open longitudinally. For further details see Bröskamp et al. [20].

First experimental results for coatings inside narrow tubes with specific temperature gradients showed an unexpected peak at a penetration depth of around 5 cm with up to 500 % higher film thicknesses. Optical inspection showed a warped and cracked layer within this region, while the thin-film seems to be intact for the rest of the tube (see Figure 1e). A reasonable explanation is the development of an intrinsic tension of vapor deposited PPX-N [42] and therefore there is potential for bulging or detachment of the thin-film from the tube surface after unloading from the temperature seesaw. In order to avoid this effect, we found that a constant temperature, e.g., near room temperature, must be applied along the tube before returning to normal pressure.

## 4. Results and Discussion

In this study, we first investigated the PPX-N deposition in narrow, one-sided closed polyamide tubes. The tube geometry is given by a length of $L_t = 20$ cm, an inner diameter of $2R_t = 2$ mm and a wall thickness of 1 mm. First experiments with a constant surface temperature or sticking coefficient, respectively, were performed in order to understand the deposition processes in detail. Later, the homogeneous layer thicknesses were realized in these tubes. All experiments were performed at an argon base pressure of 10 Pa and a precursor mass of 5 g was evaporated over 15 min with an average monomer partial pressure of 2 Pa. With a tube aspect ratio of 100 and a gas rarefaction parameter between 20 and 40, the gas dynamic regime is characterized as a slip–flow regime (see Section 2.2). Furthermore, the temperature dependence of the relative sticking coefficient was examined.

### 4.1. Thin-Film Deposition in Narrow Tubes at Constant Surface Temperature

The temperature along the temperature seesaw and tubes was set to 30 °C by means of thermoelectric elements for this experiment. The layer thicknesses along the tube were measured by thin-film interferometry and are shown in Figure 2a (black empty squares). The film thickness at the inlet of the tube is $d(0) = 2.55(6)$ μm, which is about the same as the film thickness on a reference slide (placed at a similar position in the reactor) and decreases by 90% to roughly 300 nm at the tubes closing end. The choice of the evaporated precursor amount of 5 g, and the associated high layer thickness at the inlet of the tube, permitted high accuracy for the measurement by means of thin-film interferometry over the entire distance (occurrence of maxima and minima from a layer thickness of 200 nm in the reflectance spectrum may be observed). The $\alpha$ factor was determined by fitting Equations (10) and (11) to 8.29(14) and 8.26(17), corresponding to whether we took into

account or neglected the deposition at the tubes end, respectively. The similarity of the fitting values for both equations is due to the high aspect ratio of $Ar = 200$, which shows that the deposition at the closing surface relative to the total deposited PPX-N volume inside the tube may be neglected. Further, at first glance the simplification of Equation (14) in the form of Equation (11) predicts the experiment in an excellent manner (see Figure 2a red line), pointing out that the complex terms in the mathematical description of PPX-N deposition in tubes may be neglected.

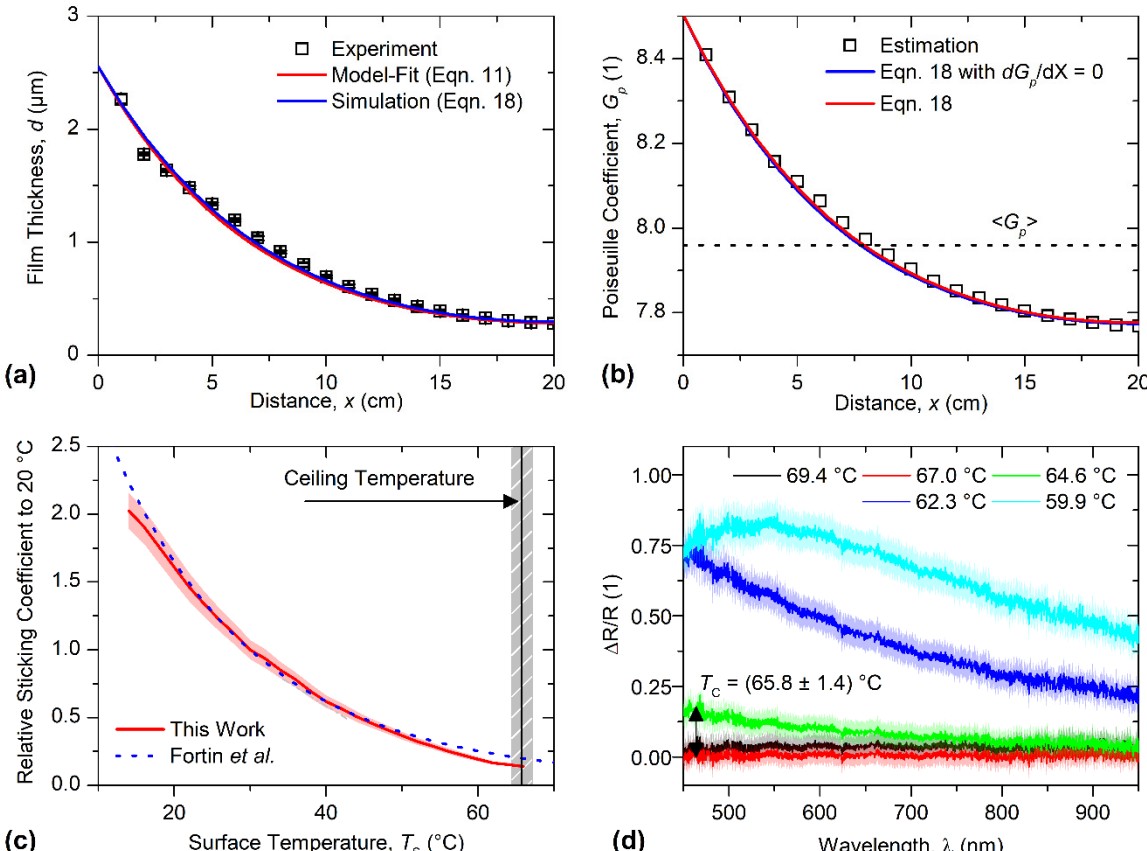

**Figure 2.** (**a**) Measured PPX-N film thickness along the tube for the condition of constant temperature (black squares). The red line results from the fit of Equation (11), while the blue line corresponds to the numerical solution of Equation (18). (**b**) Estimated Poiseuille coefficient from the spatial film thickness trend (black squares) and its mean value (black dashed line). The blue and red lines correspond to the numerical solution of Equation (18), with the blue line excluding the term containing $dG_p/dX$. (**c**) The determined relative sticking coefficient of PPX-N with respect to a surface temperature of 30 °C is represented by the red line. A fit of the model of Fortin et al. to the data is shown as the blue dashed line (see Equation (19)). For comparison the measured ceiling temperature is also shown. (**d**) Relative change in optical reflectance of a coated slide at different temperatures compared to an uncoated slide. The ceiling temperature is determined to be between the highest temperature with $\Delta R/R \neq 0$ and the lowest temperature with $\Delta R/R = 0$.

According to Equation (9), and the definition of the loss coefficient

$$\alpha = \frac{2}{\sqrt{\pi}} \frac{L_t^2}{R_t^2} \frac{S(T_{\text{ref}})}{\langle G_p \rangle} \tag{17}$$

the absolute sticking coefficient at our reference temperature of $T_{\text{ref}} = 30$ °C may be obtained. An analysis of the boundary value problem of Equation (13) should also provide deeper insight into the containing terms and the possibility of reducing the complexity until the simple form of Equation (9) is reached. Since the film growth rate is directly

proportional to the pressure and sticking coefficient, with the latter being constant in this experiment, the mean Poiseuille coefficient may firstly be estimated. For this, the spatial pressure trend inside the tube during the experiment was determined by multiplying the measured relative film thickness by the average monomer partial pressure at the inlet of the tube. The mean pressure at the inlet was known from the capacitance manometer measurement (see Figure 1d) and the value was calculated as $\langle p_M(0) \rangle = 1.65$ Pa for a process time of approximately 17 min. By plugging in all known physical parameters, namely the scattering cross sections, into Equation (16), the Poiseuille coefficient is given by $G_p(X) = 1 + 0.49 \, \text{Pa}^{-1}(D(x)p_M(0) + 1.15 \, p_{tot})$. The results from our experimental estimation are given in Figure 2b, with black squares. As previously stated, with a rarefaction parameter of $\delta = 20 \ldots 40$, the Poiseuille coefficient is expected to be somewhat between 6 and 11. Our estimated values, between 7.8 and 8.8 lie perfectly within this range, and eventually a spatial and temporal mean value of $\langle G_p \rangle \approx 7.9$ is calculated. With this, the absolute sticking coefficient at our reference temperature is given by $(T_{ref}) \approx 1.4 \times 10^{-3}$. This value shows a high discrepancy to the quantity reported by Fortin et al., which is $S(T_{ref}) = 6.3 \times 10^{-5}$ [23], and may be attributed to the energy barrier for adsorption on different materials. However, our method yields reliable results, since it is known that the sticking coefficient is in the range of $10^{-3}$ at room temperature, and therefore this value will be used for further simulations.

Thus, Equation (13) can be transformed to incorporate experimentally obtained data. In particular, the determined $\alpha$ should replace the absolute sticking coefficient, so that only the relative change of the sticking coefficient with $S_r(T_s) = S(T_s)/S(T_{ref})$ must be known for later temperature variations. This is given by

$$\frac{d^2 P}{dX^2} + \left[\frac{1}{G_p}\frac{dG_p}{dX} - \frac{1}{2\,T_g}\frac{dT_g}{dX}\right]\frac{dP}{dX} - \alpha\,\frac{G_p}{G_p}S_r(T_s)P = 0, \quad \frac{dP}{dX}\Big|_1 = -\frac{\alpha}{2\,Ar}\frac{G_p}{G_p(1)}S_r(1)P(1), \; P(0) = 1 \tag{18}$$

The solution of the boundary value problem for the constant temperature $T_{ref}$ and $\alpha = 8.29$ is shown in Figure 2a (blue line). The difference to the rather straightforward solution of Equation (11) (red line) is marginal. Since for constant temperatures the term in Equation (18) with the temperature gradient is zero, this suggests that the local change in the Poiseuille coefficient appears to be negligible. An analysis of the relative changes in the quantities shows that the pressure (direct mapping by film thickness) decreases by 90% from the inlet to the end, while the Poiseuille coefficient only changes (decreases) by less than 10%. Further, Figure 2b (blue and red lines) shows the variation of $G_p$ with the local solution for the pressure from Equation (18). At first glance, the experimental approximation agrees very well with the calculations—which is not surprising, if the quasi-equal film thickness curves of the experiment, Equation (18) and Equation (11) are considered. Furthermore, the term containing the local variation of the Poiseuille coefficient has virtually no influence on the pressure curve, as seen from the comparison of the red and blue line in Figure 2b.

### 4.2. Measurement of the Temperature Dependence of the PPX-N Sticking Coefficient

Next, the temperature dependence of the sticking coefficient was measured. The deposition experiments were obtained on soda lime glass microscope slides clamped on an aluminum slab. The surface temperature was linear between 10 °C and 70 °C at two distinct points. A temperature increment of 2 °C within this temperature interval was chosen, related to a spatial separation of thin-film interferometry measurement points on the substrates of 0.5 cm. When determining the sticking coefficient, it is clearly more convenient to specify the behavior relative to a given reference temperature. Thus, the pre-factor $(1 - \Theta)$ was eliminated in the Langmuir isotherm description of $S(T_s)$, which is given by the product of a temperature-dependent function and the fraction of reactive surface cites via $S(T_s) = S_0(T_s)(1 - \Theta)$ [23]. Furthermore, the determination of the relative sticking coefficient was sufficient for our problem due to Equation (18), and a detailed investigation of the deposition kinetics is not aimed at in this work. Figure 2c shows

the evaluated relative sticking coefficient $S_r(T) = S_0(T_s)/S_0(T_{ref})$, corresponding to a reference surface temperature of $T_{ref} = 30\,°C$ (therefore $S_r(30\,°C) = 1$ holds by definition). The data are given in Table 1 and depicted in Figure 2c (red line) with an almost exponential course of the relative sticking coefficient within our investigated temperatures. Above the ceiling temperature $T_c$ (see below for a precise identification), no film growth takes place. Just below the ceiling temperature, the relative sticking coefficient was determined to be $S_r = 0.14(1)$, which then falls instantly to zero when exceeding $T_c$. Here, a smooth transition with a continuously decaying $S_r(T_s)$ is not found, as is the case in several deposition models [22,23]. The minimum temperature in this study was 14 °C with $S_r = 2.02(13)$. Thus, for the specification of the temperature gradient for the formation of a homogeneous PPX-N layer thickness inside narrow tubes, a maximum variation of the sticking coefficient by a factor of 15 was granted when working within our investigated temperature range. To compare our measured data with the Fortin model, a nonlinear curve regression was performed. The model is given with [23]

$$S(T_s) = (1 - \Theta)\, S_0(T_s) = (1 - \Theta)\frac{1}{1 + V \exp(-\Delta E/RT_s)} \tag{19}$$

where $\Delta E = E_d - E_a$ are the energy barriers to desorb $E_d$ and chemisorb $E_a$, $V$ a probability constant and $R$ the universal gas constant. The curve fit is shown in Figure 2a by the blue dotted line. We found the energy difference $\Delta E = 41.37\ kJ\ mol^{-1}$ being in perfect agreement with Fortin's value of $\Delta E = 39.4\ kJ\ mol^{-1}$. The probability constant is, in our case, $V = 1.16 \times 10^8$, where Fortin et al. report $V = 1.2 \times 10^8$. For the absolute sticking coefficient at reference temperature an estimate can be obtained from fitting the loss coefficient $\alpha$ to the constant temperature tube experiments. We found that $S_0(T_{ref})(1 - \theta) \approx 0.44(9) \times 10^{-3}$ for our experimental data (see above) which deviates from the value calculated from the Fortin model.

**Table 1.** Measured relative sticking coefficient at 30 °C, over a wide range of varying surface temperature ($\Delta T_s \approx 50\,°C$) up to the ceiling temperature. Temperature $T_s$ is given in units of °C and uncertainties are in brackets.

| $T_s$ | $S_r$ | $T_s$ | $S_r$ | $T_s$ | $S_r$ |
|---|---|---|---|---|---|
| 14 | 2.02(13) | 32 | 0.93(07) | 50 | 0.37(03) |
| 16 | 1.91(13) | 34 | 0.85(06) | 52 | 0.32(03) |
| 18 | 1.76(12) | 36 | 0.78(06) | 54 | 0.29(02) |
| 20 | 1.61(11) | 38 | 0.69(06) | 56 | 0.25(02) |
| 22 | 1.46(10) | 40 | 0.62(04) | 58 | 0.22(02) |
| 24 | 1.33(09) | 42 | 0.57(04) | 60 | 0.19(02) |
| 26 | 1.22(09) | 44 | 0.51(04) | 62 | 0.16(01) |
| 28 | 1.12(08) | 46 | 0.46(03) | 64 | 0.15(01) |
| 30 | 1.00(07) | 48 | 0.41(03) | 66 | 0.14(01) |

Although the model parameters are in good agreement, an increasing discrepancy towards the ceiling temperature between model and experiment becomes apparent. This mismatch is attributed to the insufficient accuracy of the chemisorption model for temperatures close to the ceiling temperature [10] (see also $S_0(T_s)$, which falls off continuously in contrast to the observed abrupt drop). As partial suppression of deposition at the inlet of the tube is necessary for a homogeneous coating, this temperature range remains of crucial interest for our work. Therefore, we use a smoothing spline to our experimental data instead of the chemisorption model for further calculations.

Due to discrepant literature values, the ceiling temperature was further resolved using thin-film interferometry. To do so, we calculated the relative change in reflectance of the coated substrate compared to a bare soda lime microscope slide, which is given by $\Delta R/R = (R_{coated} - R_{uncoated})/R_{uncoated}$. Thus, in the absence of a PPX-N thin film on the substrate, $\Delta R/R$ will remain zero over the entire spectral range (400 nm to 1000 nm). For

selected temperatures around the ceiling temperature, the measured reflectance spectra are shown in Figure 2d. At the lowest temperature of 59.9 °C a maximum at about 550 nm is observed, but for the other temperatures is not. This is equivalent to the behavior known from thin-film interferometry, where with increasing film thickness the number of maxima as well as minima grows ($R \propto \cos(nd/\lambda_{\mathrm{p}})$ with $n$ the refractive index of the thin-film and $\lambda_{\mathrm{p}}$ the photon wavelength [20,43]. Consequently, this behavior is observed with decreasing surface temperatures (not shown here). The ceiling temperature is determined by taking the average of the lowest temperature with $\Delta R/R = 0$ and the highest temperature with $\Delta R/R \neq 0$. Thus, we can report the ceiling temperature of PPX-N on soda-lime glass as 65.8(1.4) °C in the range of an average monomer partial pressure of approximately 2 Pa along 10 Pa argon base pressure. This value is in good agreement with our previous measurements with 70(2) °C [20], but strongly contrasts the values published by Yang et al. and Fortin and Lu [10,23–25], which are 30 °C and somewhat above 40 °C, respectively. We consider that the approach of determining $T_{\mathrm{c}}$ by linear extrapolation of deposition rates from room temperature and below room temperature experiments causes the large discrepancies, since the actual, rather exponential, decay of $S_0(T_{\mathrm{s}})$ is greatly overestimated within this method.

### 4.3. Defining the Required Temperature Gradient and Boundary Value Problem Analysis

Next the temperature gradient for the generation of homogeneous layer thicknesses was investigated. For this, according to Equation (1), the product of local sticking coefficient and monomer partial pressure must remain constant within the entire tube. $S(X)P(X) = S(0)P(0) = S(0)$ applies. A natural consequence is that the resulting pressure gradient compared to a non-manipulated process is reduced, and therefore a homogeneous layer thickness may only be realized by a partial suppression of deposition on the first section of the tube and vice versa. Owing to the complexities of Equation (14) and Equation (18), Equation (9) was used to approach this problem. The disregard of complex and spatially varying terms has already been discussed in the previous section, with the result that a simplified mathematical description of the problem yields sufficiently accurate results—at least when dealing with constant temperatures. With the given domain condition $S(X)P(X) = S(0)$, Equation (9) yields a parabolic pressure profile after twofold integration

$$P(X) = \alpha \, S_{\mathrm{r}}(0) \left[ \frac{X^2}{2} - \left( 1 + \frac{R_{\mathrm{t}}}{2\,L_{\mathrm{t}}} \right) X \right] + 1 \tag{20}$$

$\alpha$, determined by fitting Equation (11), already includes the absolute sticking coefficient for the constant reference temperature $T_{\mathrm{ref}}$. This makes it sufficient, as mentioned in the previous section, to calculate the sticking coefficient relative to the temperature present in the original experiment. The parabolic course of the pressure gradient follows with $P(X) > 0$ for $0 < X < 1$ an upper bound for the applicable sticking coefficient within this model,

$$S_{\mathrm{r}}(0) < \frac{2}{\alpha \, (1 + R_{\mathrm{t}}/L_{\mathrm{t}})} \tag{21}$$

where, for long aspect ratios, this is simplified to $S_{\mathrm{r}}(0) < 2/\alpha$. From the measurement of the sticking coefficient, we further know that the minimum relative sticking coefficient (just below the ceiling temperature) is $\min\{S_{\mathrm{r}}\} = 0.14$. By defining a new relative and ideal film thickness via $D_{\mathrm{r}}(X) = D_{\mathrm{manipulated}}/D_{\mathrm{unmanipulated}} = S_{\mathrm{r}}(X)\,P(X) = S_{\mathrm{r}}(0)$ we find a small theoretical bandwidth for achieving homogeneous film thicknesses with values between 0.14 and 0.24 $d(0)$. Here $d(0)$ is the film thickness at constant reference temperature at the inlet of the tube, which is 2.55(6) μm (see above).

The required sticking coefficient profile along the tube is then finally given with

$$S_{\mathrm{r}}(X) = \left[ \alpha \left[ \frac{X^2}{2} - \left( 1 + \frac{R}{2\,L} \right) X \right] + 1/S_{\mathrm{r}}(0) \right]^{-1} \tag{22}$$

In particular, Equation (22) is valid for manipulated CVD processes in tubes with linear deposition kinetics, and the corresponding temperature gradient is derived from the specific dependence of the sticking coefficient. In our case, we will use for the final experiments the numerical smoothing spline from our measured relative sticking coefficient.

At this point, the employed approximation shall be discussed. In this regard Equations (9) and (18) were solved numerically using the defined sticking coefficient from Equation (22). Since we know $L_t \gg R_t$ the aspect ratio in the equation was neglected. The calculation was carried out in Comsol Multiphysics 5.5, and the required sticking coefficient was defined for a constant film thickness corresponding to 0.9 times the maximum achievable homogeneous film thickness (Equation 21) (see Figure 3a). With $\alpha = 8.29$ for our tube geometry at a reference temperature of 30 °C, the relative film thickness corresponds to 21.7 % of the film thickness at the beginning of the tube of a non-manipulated process. This ratio was calculated as $D_r(X) = D_{\text{manipulated}} / D_{\text{unmanipulated}} = S_r(X)\,P(X)$. Figure 3b shows the determined pressure curves $P(X)$ in comparison. Here, the blue dashed line depicts the solution with the assumption of a constant Poiseuille coefficient as well as constant gas temperature, according to Equation (9). The red curve shows the pressure solution for the boundary value problem from Equation (18), where the variable coefficient $G_p$ and the local temperature change of the gas along the tube ($T_g = T_s$) were taken into account. The latter was determined from the model of Fortin et al. via the relative sticking coefficient. This was necessary since the absolute values at the reference temperature of 30 °C differ from Fortin et al. and our work. A comparison of the two solutions (red and blue line in Figure 3b) does not reveal any differences at first glance. However, for both calculations, the pressure gradient turns out to be lower in comparison to a deposition process at constant temperature (black dashed line, corresponding to solution from Figure 2a). In order to investigate the virtually non-existent mismatch between the simple and complex form of the boundary value problem, the corresponding terms of the differential equation (Equation (18)) are shown in Figure 2c. By definition, the sum of these must equal zero, which corresponds to the black dashed curve. The increased complexity of Equation (18) is caused by the local temperature dependence of the gas as well as a spatially variable Poiseuille coefficient, which are part of the term containing the first pressure derivative $dP/dX$. This contribution is shown as a magenta line in Figure 3b and obviously yields the smallest influence on the overall solution. The terms with the second pressure derivative $d^2P/dX^2$ (red line) as well as the pressure $P$ itself (blue line) are thus approximately axially symmetric around the zero line (corresponding to the difference of the magenta curve). This explains the negligible deviation of the solutions of Equations (9) and (18) in Figure 3b and supports the assumption that, for the derivation of the required sticking coefficient, it is sufficient to consider the simple form of the differential equation.

Finally, the pressures from Figure 3b for the simple as well as complex differential equation can be multiplied by the relative sticking coefficient in order to obtain the relative layer thickness $D_r(X)$. The results are presented in Figure 3d as difference to the ideal value according to the following expression

$$\frac{\Delta D_r(X)}{D_r(0)} = \frac{D_r(X) - S_r(0)}{S_r(0)} \tag{23}$$

Here, the black dashed line corresponds to the pressure solution from Equation (9) and, as required by theory, gives a perfect match with the zero line. By choosing the representation with the relative layer thickness change, individual contributions in the differential equation can be examined more precisely. Especially those which may not be clearly evident in the pressure course. Therefore, individual contributions were systematically exhibited in Equation (18), namely the temperature dependence of the gas (blue line) and the local dependence of the Poiseuille coefficient (red line). The solution for considering all terms (red pressure plot from Figure 3b) is shown in magenta, and the dashed lines refer to results derived from neglecting the deposition at the closed end of the tube. At first glance, it can be seen that, by forcing the pressure gradient to zero at the end

of the tube, the monomer partial pressure in this region increases slightly, which exerts a high effect on the film thickness profile due to the high relative sticking coefficient at this position (above two, see Figure 3a). Furthermore, neglecting the temperature gradient as well as the Poiseuille coefficient gradient, ergo the term in Equation (18) which contains the pressure derivative, leads to similar results as considering the entire terms, especially in the last third. The reason for this is the opposite action of the temperature gradient and the Poiseuille coefficient gradient, which have a different sign in the $dP/dX$ term in Equation (18). Since the relative changes in these two quantities are of the same order of magnitude, as discussed in the theory section, considering the two quantities separately in the BVP has opposite effects, which can be seen in the comparison of the red and blue graphs in Figure 3d. Here, the change in relative layer thickness is in the range of plus minus four percent. However, the overall moderate deviation of a few percent from the zero line when examining different contributions reveals that the basic form of the boundary value problem according to Equation (9) provides an excellent approximation. Furthermore, it should be pointed out that the exact course of the gas temperature along the tube is not known and a detailed investigation is not within the scope of this work, and it is clear from the differentiated consideration that relative deviations from theory and experiment of a few percent may arise.

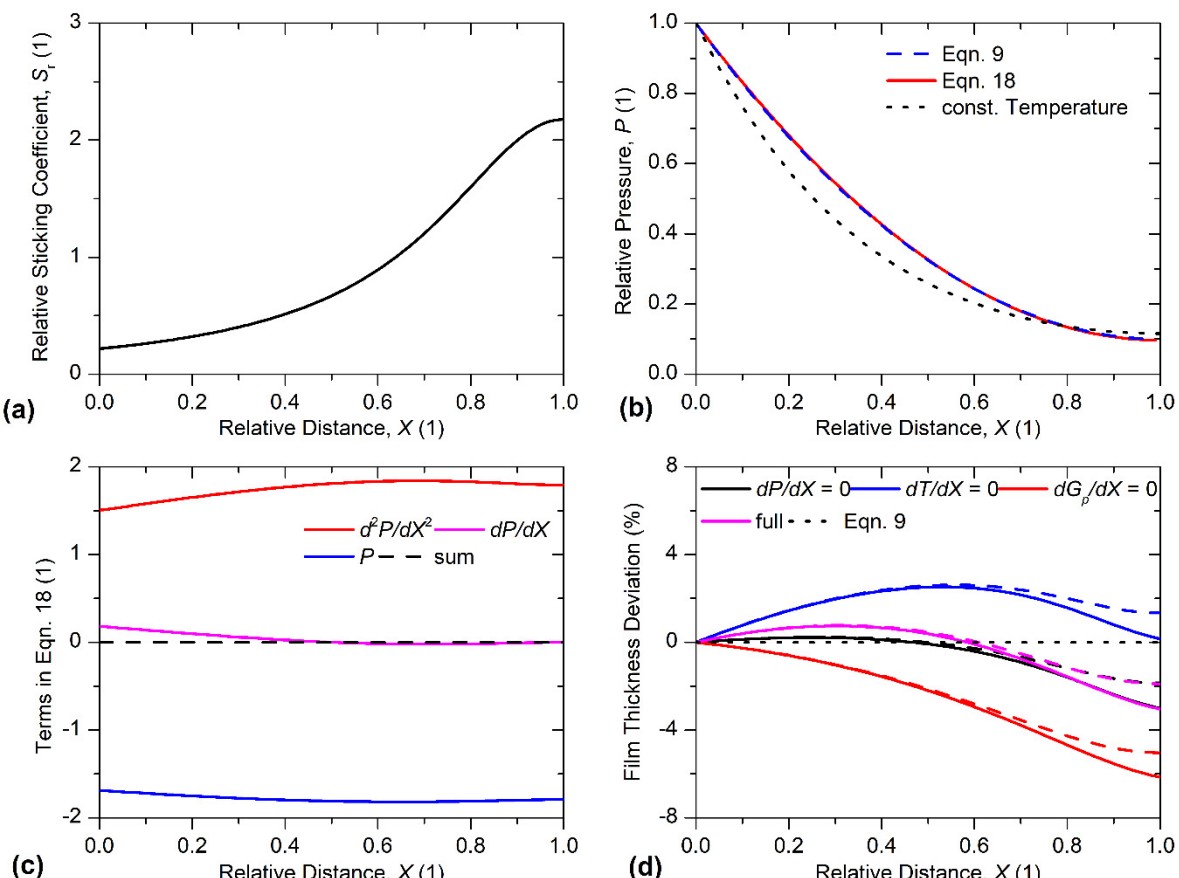

**Figure 3.** (**a**) Derived theoretical relative sticking coefficient for a homogeneous coating of $0.9 \cdot 2/\alpha = 21.7\%$. (**b**) Pressure solution for boundary value problem containing all contributions (red curve, Equation (18)) and the basic form with all parameters set constant (blue dashed line from Equation (9)). The pressure curve for a constant temperature profile is shown as black dotted line. (**c**) Graphical representation of the terms appearing in Equation (18). By definition, the sum of all colored curves corresponds to the zero line (black dashed line). It becomes evident that the term containing the pressure derivative has the least impact on the total solution. (**d**) Deviation of the

relative film thickness to the theoretical value $\Delta D_r(X)/D_r(0)$ calculated for various contributions within Equation (18) which are given in colored lines. The black dotted line is the solution for Equation (9) which is zero by definition, and dashed colored lines represent the suppression of film growth at the tubes end. It becomes evident that neglecting the term containing $dP/dX$ in Equation (18) (black line) yields comparable results as considering all terms in Equation (18) (magenta line). This is due to the opposite action of temperature gradient (red) and Poiseuille coefficient gradient (blue).

### 4.4. Deposition of Homogeneous PPX-N Thin-Films Inside Narrow Tubes

Finally, a sticking coefficient curve for the generation of a homogeneous PPX-N film thickness for our tubes with $\alpha = 8.29$ was calculated according to Equation (22). The resulting temperature reference points for the temperature control were determined from the measurements of the relative sticking coefficient. As stated in Section 3, the exact temperature profile or sticking coefficient profile within the tube is not known, and is only accessible via a heat conduction equation (here using Comsol Multiphysics). The predicted temperature curve can in turn be transformed into a real sticking coefficient profile. This is shown for three experiments with slightly varying temperatures in Figure 4 (upper row) as a blue line. Here, the center column is to be considered as a reference, whereas the left or right course exhibits slight temperature differences at certain reference points. The temperatures measured from the experiment (at just these five reference points) were also converted into a sticking coefficient and are shown as empty black diamonds in Figure 4. A slight deviation from the blue curve, especially at the last reference point at 18 cm, can be seen. In particular, there are plateaus at the beginning and end of the tube for the sticking coefficients, which are caused by the limited geometry of the experimental setup. Thus, it should be noted that it is technically demanding to map the required theoretical sticking coefficient exactly along a long tube.

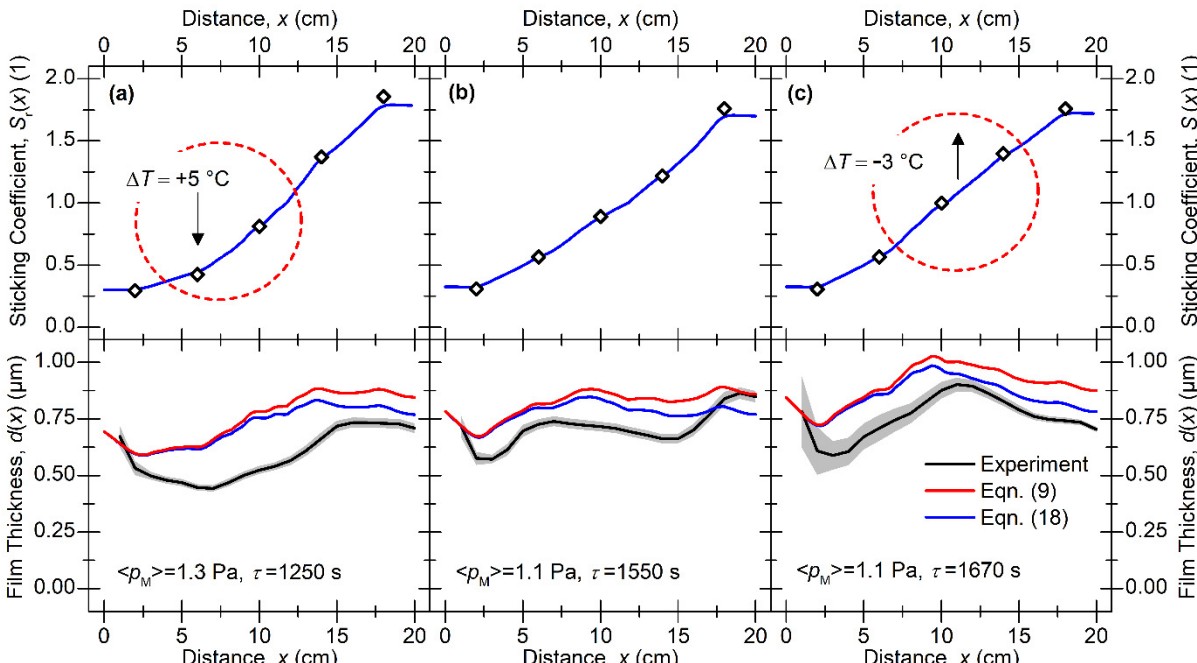

**Figure 4.** Upper row relative sticking coefficient vs. penetration distance $x$. The black dashed line is ideal sticking coefficient. The blue line is the sticking coefficient calculated from the local temperature field and temperature measurement points are black diamonds. The lower row shows the film thickness with black line measurement, the blue line shows the simulation with all contributions (Equation (18)) and the red line shows the simple form of the boundary value problem (Equation (9)).

The resulting film thicknesses (process parameters as at constant temperature), corresponding to the applied temperature fields, are shown in Figure 4 as black lines in the

bottom row. At first glance, it appears that all three temperature curves lead to homogeneous layer thicknesses with corresponding accuracy. The relative deviation from the mean value is no more than 20 % for all three experiments, and higher film thicknesses could even be obtained at the end of the tube when compared to the inlet. If the film thickness from the center column is considered as a reference, the slight temperature variations in the other experiments and their effect on the film thickness may be discussed. If the left experiment is taken into account, where the second and third temperature is increased by 5 °C, a layer thickness decrease of about 20–30% is found at this point. When examining the sticking coefficient at the second reference point, which is 47.4 °C on the left and 42 °C in the center, the sticking coefficient for the higher temperature is reduced by 20–25%. The same applies to the third temperature point. A comparison with the right experiment shows the contrary trend. Here, the temperatures at the third and fourth support points are reduced by 3 °C. This corresponds to an increase of the sticking coefficient by 10 and 20% (third and fourth point, respectively). The layer thickness increases correspondingly by approximately 15%, with a pronounced maximum in the center of the tube. Thus, it is straightforward to demonstrate that the expected change in film thickness within the tube approximately matches the relative change in the sticking coefficient.

Using the specified sticking coefficient curves, the film thicknesses along the tubes were simulated. The solutions according to Equations (18) and (19) which correspond to the consideration of all terms and the simplified form of the boundary value problem, respectively, were examined. The simulation of the monomer partial pressure was carried out analogously to the previous subsection. From this, the real film thickness can be calculated from the product of the local deposition rate according to Equation (1), multiplied by the corresponding process time. This is written as

$$d_{\mathrm{f}} = \frac{\langle p_{\mathrm{M}} \rangle \tau}{\sqrt{\pi}\, \hat{c}_{\mathrm{g}}\, \rho_{\mathrm{f}}}\, S(X)\, P(X) \tag{24}$$

with the density of the PPX-N layer $\rho_{\mathrm{f}} = 1.1$ g/cc [41]. The solutions of the simulation are given in the lower row of Figure 4, where the red curve belongs to the boundary value problem of Equation (9) and the blue curve includes all variable terms. The process parameters such as mean pressure $\langle p_{\mathrm{M}} \rangle$ at the inlet of the tube and the process duration $\tau$ are also given in the bottom row. It should be noted that initial results deviated more from the experiments, so we reduced the absolute value of the sticking coefficient by a factor of 1.75 for better comparison. Through this, the spatial response of experiment and simulation is in good agreement for all temperature gradients. Further, to better match the absolute value of the film thickness, we multiplied the pressure at the inlet of the tube by a factor of 0.7 in the simulation. As a result, the modeled absolute layer thicknesses only deviate from the experiment by a factor no larger than 1.2. It should be noted, however, that despite all approximations in the derivation of the boundary value problem, the simulations hit the same order of magnitude from the experiment without appropriate corrections. In general, we find a very good agreement between experiment and theory and distinctive points, such as the maximum layer thickness in the center of the tube of the right experiment, can be reproduced very well. Furthermore, the sloped bend at the inlet of the tube is correctly reproduced for all three experiments. We found that the origin of this is the finite size of the thermoelectric elements, and thus the sticking coefficient can only be mapped along the tube with limited accuracy. A comparison of the blue and red curves in Figure 4 also shows that the simple form of the DGL already gives good results. Especially in the first third of the tube, which is in good agreement with the analysis in the previous subsection. However, it can be seen that the full consideration of all terms leads to a better agreement between experiment and theory. In particular, it is shown that the simple form of Equation (9) somewhat overestimates the layer thickness in the second half of the tube. The relative deviations to each other, however, amount to only 5–10%, which is in good agreement with the analysis on the contributions of the individual terms in the previous subsection. Thus, it can be concluded that the detailed knowledge about the process parameters as well as

the absolute sticking coefficient are rather more important than the inclusion of further correction terms in the physical modeling.

## 5. Summary and Conclusions

In this work, a formal basis for the generation of longitudinal homogeneous film thicknesses in narrow tubes, formed in a CVD process, is presented. As example, the Gorham process for the deposition of poly-para-xylylene (PPX-N) thin-films was considered. From the steady state mass flow equation in tubes, a generalized boundary value problem was derived yielding established models through corresponding approximations. Furthermore, the boundary problem was discussed for the special case of intermediately diluted gas mixtures which are required for a good conformity of the coating process. In order to achieve uniform layer thicknesses, a model that calculates the required local sticking coefficient was presented. As main result, a longitudinal homogeneous film can only be realized by partial suppression of the deposition at the beginning of the tube and vice versa. Thus, the gradient of the partial pressure of the reactant in the gas phase along the tube is less for a manipulated process than for deposition at constant temperature. This defines an upper limit on the relative film thickness of a manipulated process, which is referred to as the maximum deposition efficiency. For long aspect ratios, $D < 2/\alpha$ (with $\alpha$ of the loss constant determined from experiments at constant temperature). An examination of the derived differential equation shows that neglecting complex terms distorts the theoretical result by only a few percent. This ensures that the applied theory can be kept as simple as possible. Furthermore, the relative sticking coefficient was accurately measured from room temperature up to the ceiling temperature where a deviation from a chemisorption model according to Fortin et al. was observed at temperatures just below the ceiling temperature. We report an experimentally determined value of 68 °C for the ceiling temperature which is in good agreement with a recently published value, but shows a larger deviation from quantities found in literature. Finally, by predicting a sticking coefficient curve, a temperature profile for our experimental setup for the so-called temperature seesaw and nearly homogeneous PPX-N film thicknesses in thin tubes of an aspect ratio of 200 were produced. The relative deviation along the entire tube in these experiments was less than 20% of the mean film thickness, with partially higher thicknesses at the end of the tube compared to the inlet. The experimental results match the modeled differential equation well. Due to the conception of this work, the given methodology is, in principle, independent of the reactant in the gas phase and can be adapted to various processes with the knowledge of a mean temperature-dependent sticking coefficient. We further expect our method to be applicable for related processes because our monomer resembles the simplest chain former with two radical functions, which are $CH_2 - CH_2$, also known as ethylene. Therefore, PPX-N CVD may be seen as a prototype for polymerization, especially forced surface polymerization producing filamentous polymers.

**Author Contributions:** Conceptualization, D.R.; methodology, D.R. and M.B.; validation, D.R., M.B. and G.F.; formal analysis, D.R.; investigation, D.R. and M.B.; resources, G.F.; software D.R. and M.B.; writing—original draft preparation, D.R.; writing—review and editing, M.B. and G.F.; visualization, D.R.; supervision, G.F.; project administration, G.F.; funding acquisition, G.F. All authors have read and agreed to the published version of the manuscript.

**Funding:** This research was funded by the Federal Secretary of Economy, grant number KF2527-5103 CR4.

**Institutional Review Board Statement:** Not applicable.

**Informed Consent Statement:** Not applicable.

**Data Availability Statement:** The data on which the figures in this paper are based are available on request from the corresponding author.

**Acknowledgments:** The authors wish to thank Armin Hadzimujic for his technical assistance in fabricating the temperature seesaw.

**Conflicts of Interest:** The authors declare no conflict of interest.

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
