# Peer review of "Chemical Vapor Deposition of Longitudinal Homogeneous Parylene Thin-Films inside Narrow Tubes"

_processes, doi:10.3390/pr10101982_

Round 1

Reviewer 1 Report

The paper reports about the limits of the CVD processes during the production of longitudinally homogeneous coatings inside narrow tubes and find the condition to obtain uniform coatings both experimentally and through simulations. The paper is very detailed and well written. My only concern is on how specific their study is for the investigated PPX-N deposition.

The authors comment ot the end of the conclusions section that the study can be generalized to other reactants. Can they specify to which types of reactants and processes this study can be generalized? In addition what is the effect of the reactor geometry and specifically of the gas source geometry?

Reviewer 2 Report

 "Chemical Vapor Deposition of Longitudinal Homogeneous Parylene Thin-Films inside Narrow Tubes”, Article reference: process-1913210

 This paper discussed in theory and the provided model is verified by experiments. The research’s prediction of a required sticking coefficient curve yields experimentally measured homogeneous film thicknesses and shows a good agreement with the given prognosis. However, the authors should consider the amendments and should comply with the points listed below to correct the paper.

 REFEREE REPORT:

 1.     In line 31, “Arrhenius and the law of mass action, chemical reactions always depend severely on temperature, and manufacturing is challenging with respect to process parameters and their stability….”, please provide relevant references to support these claims.

2.     In line 108, sticking coefficient plays a very important role in this study, but there is a lack of references to describe the important factors that affect the sticking coefficient, please provide relevant references to describe the sticking coefficient.

3.     In equation (1), (3), and (4), The two parameters pM, Tg lack explanation, please add.

4.     In line 242, the pyrolysis temperature of dimer depends on the temperature required for pyrolysis of carbon chains, which is generally 650~680 degrees Celsius. The temperature setting in this study seems to be too low?
